# Aqueous Solution of Ionic Liquid Is an Efficient Substituting Solvent System for the Extraction of Alginate from *Sargassum tenerrimum*

**Kinjal Moradiya** [1,2], **Matheus M. Pereira** [3] **and Kamalesh Prasad** [1,2,*]

[1] Natural Products & Green Chemistry Division, CSIR-Central Salt, and Marine Chemicals, Research Institute, Gijubhai Badheka Marg, Bhavnagar 364002, India; kinjalmoradiya7@gmail.com

[2] Academy of Scientific and Innovative Research (AcSIR), Ghaziabad 201002, India

[3] CERES, Department of Chemical Engineering, Rua Sílvio Lima, University of Coimbra, Pólo II—Pinhal de Marrocos, 3030-790 Coimbra, Portugal; matheus@eq.uc.pt

[*] Correspondence: kamlesh@csmcri.res.in

**Abstract:** Three ionic liquids (ILs) and three deep eutectic solvents (DESs) with identical counterparts, as well as their aqueous solutions, were prepared for the selective extraction of alginate from *Sargassum tenerrimum*, a brown seaweed. It was found that the ILs and their hydrated systems were only able to extract alginate from the seaweed directly, while the DESs were not, as confirmed by molecular docking studies. When the quality of the polysaccharide was compared to that produced using the hydrated IL system with the widely used conventional method, it was discovered that the physicochemical and rheological characteristics of the alginate produced using the ILs as solvents were on par with those produced using the conventional method. The ILs can be seen as acceptable alternative solvents for the simple extraction of the polysaccharide straight from the seaweed given the consistency of the extraction procedure used in conventional extraction processes. The hydrated ILs were discovered to be more effective than their non-hydrated counterparts. The yield was also maximized up to 54%, which is much more than that obtained using a traditional approach. Moreover, the ionic liquids can also be recovered and reused for the extraction process. Additionally, any residual material remaining after the extraction process was converted into cellulose, making the process environmentally friendly and sustainable.

**Keywords:** ionic liquids; deep eutectic solvents; extraction; seaweed; choline alginate; sodium alginate





## 1. Introduction

Alginates, versatile biopolymers, hold immense promise as biodegradable and renewable materials. Their extensive applications span diverse industries, such as the pharmaceuticals, food, textiles, cosmetics, and rubber goods manufacturing industries; industries related to the production of welding rods and paper printing dye; and various other sectors [1]. Alginates, crucial biopolymers, are sourced from brown seaweed species belonging to the Phaeophyceae class. Notably, alginates are extracted from various brown seaweed species, including *Sargassum* spp., *Cystoseira* spp., and *Turbinaria* spp. The extraction of alginates from these seaweed species highlights their significance as abundant and renewable resources for producing versatile biopolymers. These alginates play a vital role in a myriad of industrial applications owing to their unique properties and eco-friendly nature [2]. Alginate, a polysaccharide, is composed of covalently bonded D-mannuronate (M) and its epimer, L-guluronate (G), arranged in various sequences. The term "alginates" encompasses alginic acids and their salts, which include sodium, calcium, potassium, ammonium, and magnesium, commonly used in various applications [3].

Alginate exhibits a range of beneficial properties, and being nontoxic, biocompatible, and biodegradable, it is desirable in many applications where these properties play a crucial role. In the food industry, alginate serves as a coating agent, a protective barrier against microbiological threats, and a versatile agent for gelling, thickening, stabilizing, or emulsifying purposes [4]. In the biomedical field, alginate from brown seaweed demonstrates antibacterial capabilities, potent free radical scavenging, antioxidant activity, renoprotective effects, and anticancer and immunostimulatory activities. Its richness in dietary fiber makes it applicable in treatments for diabetes, liver and parathyroid diseases, and the repair and regeneration of tissues and organs. Alginate-based scaffolds are utilized for various medical conditions as a matrix for cell growth, extending to applications in the dental field and tissue engineering, including traumatic, chronic, and surgical wound healing [5].

Furthermore, alginate is integral to drug delivery systems, manifesting in diverse forms such as gels, matrices, membranes, nanospheres, and microspheres [6]. As the industrial use of alginate expands, the extraction process must align with the demands of simplicity and environmental friendliness, highlighting the importance of ensuring a straightforward and eco-conscious approach in alginate extraction procedures.

The use of ionic liquids (ILs) as an alternative to volatile organic solvents has gained significant attention. Classified as green solvents, ILs are characterized by their recyclability, nonflammability, and nonvolatility. They are considered as favorable mediums for chemical syntheses due to their exceptional properties, including their remarkable solvating potential, thermal stability, and tunability through the choice of suitable cations and anions [7–9]. These attributes make ILs versatile and environmentally friendly, positioning them as promising candidates for various applications in the field of chemical synthesis and beyond. Because of all of these properties, ILs are the much preferred choice for various applications, such as the extraction of chlorophyll [10], wastewater treatment, the production of biofuels [11], and the electrospinning of cellulose [12]. Regarding the liquid–liquid microextraction of organic compounds and metals from diverse matrices such as water, food, and cosmetics, a range of extraction techniques are employed [13], such as the extraction of bioactive molecules [14]; the extraction of seaweed polysaccharides [15]; the IL-based extraction of metal ions, including alkali, alkaline earth, heavy, and radioactive metals [16], the extraction of amino acids [17] and short-chain carboxylic acids (acetic acid, butyric acid, lactic acid) [18]; the extraction of toluene, cyclohexanone [19], phenol, tyrosol [20], and the antibiotics amoxicillin and ampicillin from aqueous solution at pH = 8 by 1-octyl-3-methylimidazoliumtetrafluoroborate [21], etc. DESs are fluids composed of two or three components that can self-assemble through hydrogen bond interactions to produce a eutectic combination with melting temperatures lower than those of the individual constituents. While DESs often become liquids at temperatures below 100 °C, it is important to note that some components may have associated risks [22]. Due to all of these properties, DESs are also useful in many fields, including in the pre-treatment of biomass polymers [23], extraction–desulfurization [24], the extraction of seaweed polysaccharides [15], biosensor development [25], the development of polymeric membranes [26], the separation of bioactive compounds from medicinal plants [27], etc. No single study in which ILs are used as a single extracting agent for alginate extraction is available. In addition to their versatile properties, both ionic liquids (ILs) and deep eutectic solvents (DESs) find widespread applications, especially in the extraction of seaweed polysaccharides and bioactive molecules. Despite their effectiveness, there is a current preference for using aqueous solutions of ILs and DESs over concentrated solutions for extraction purposes. Studies have shown that hydrated DESs exhibit superior extraction efficiency in extracting carrageenan from *Kappaphycus*, emphasizing the potential of DESs as effective solvent systems for biomass extraction and highlighting the importance of considering solvent hydration in optimization strategies for biomass fractionation [28]. Previous studies have employed Ionic Liquids (ILs) such as Tetramethylammonium hydroxide (TMAH) for the extraction of alginic acid from seaweed polysaccharides. The extraction process involves an acid treatment followed by the use of TMAH, highlighting the versatility of ILs

in polysaccharide extraction processes [29]. This study aligns with this trend, opting for environmentally friendly aqueous solutions of both solvents. The emphasis on aqueous solutions reflects a commitment to sustainability and aligns with the contemporary shift towards eco-conscious practices in solvent-based applications.

In this study, the effective selective extraction of alginates from *Sargassum tenerrimum* seaweed powder using deep eutectic solvents, bio-based ionic liquids, and their hydrated combinations is demonstrated. The bio-ILs used in this study were choline glycolate, choline acetate, choline formate, and their DES counterparts, i.e., choline chloride/glycolic acid (1:2), choline chloride/acetic acid (1:2), and choline chloride/formic acid (1:2).

## 2. Materials and Methods

### 2.1. Materials

The extraction studies in this study utilized Choline bicarbonate and Choline chloride, which were sourced from Sigma-Aldrich (Burlington, MA, USA) and Alfa-Aesar (Ward Hill, MA, USA), respectively. Sodium carbonate, acetic acid, formic acid, hydrochloric acid, 35% $w/w$ solution of hydrogen peroxide, sodium hydroxide pellets, and laboratory-grade 2-propanol (IPA) were procured from Sisco Research Laboratories Pvt. Ltd., Mumbai, India (SRL). A 70% aqueous solution of glycolic acid was obtained from Tokyo Chemical Industry Co. Ltd., Tokyo, Japan (TCI).

The brown seaweed species, *Sargassum tenerrimum*, used in this study was harvested from the southeast coast of India (9°15′ N; 78°58′ E). The seaweed was shade-dried after sun-drying, water-washed multiple times to eliminate any epiphytes and sand, and then further dried. Using a mortar and pestle with the addition of liquid nitrogen, the dried seaweed was finely crushed into a powder for further processing.

### 2.2. Synthesis and Characterization of Bio-Based Ionic Liquids and Deep Eutectic Solvents

All bio-based Ionic liquids (ILs) were synthesized using previously reported acid–base neutralization methods [Supplementary Information (Schemes S1–S3)] [30,31]. In a typical reaction, the corresponding acid was introduced in an equimolar ratio (1:1) to a round-bottom flask containing aqueous choline bicarbonate (80 wt.% in water). The reaction mixture was stirred continuously in an inert environment, and after the complete addition of each acid, the reaction was refluxed at 60 °C for 24 h. To remove any unreacted starting components, the synthesized ILs were washed with ethyl acetate. The collected ILs were then dried under reduced pressure and stored in sealed glass vials, shielded from light, and kept in a dry environment. FT-IR [Supplementary Information (Figures S5–S7)], [1]H NMR, and [13]C NMR were performed to confirm the synthesized and recovered ILs' structural (Figure 1) integrity. NMR data: [1]H NMR spectra of choline glycolate (D$_2$O, 600 MHz, $\delta$/ppm relative to TMS): 3.20 (s, 9H,-N-CH$_3$), 3.51, 3.52, 3.53 (t, 2H, -CH$_2$-N-), 4.05 (t, 2H, -O-CH$_2$-), 3.94 (t, 2H, -CO-CH$_2$-O-), 4.85 residual solvent peaks [Supplementary Information (Figure S1)]. [1]H NMR spectra of recovered choline glycolate (D$_2$O, 600 MHz, $\delta$/ppm relative to TMS): 3.18 (s, 9H, -N-CH$_3$), 3.49, 3.49, 3.50 (t, 2H, -CH$_2$-N-), 4.03, 4.02 (t, 2H, -O-CH$_2$-), 3.90 (t, 2H, -CO-CH$_2$-O-), 4.93 residual solvent peaks were observed, [Supplementary Information (Figure S4)]. [1]H NMR spectra of choline acetate (D$_2$O, 600 MHz, $\delta$/ppm relative to TMS): 1.95 (s, 3H, -CO-CH$_3$), 3.20 (s, 9H, -NCH$_3$), 3.50, 3.51, 3.52 (t, 2H, -CH$_2$-N-), 4.04, 4.05 (t, 2H, -O-CH$_2$), 5.04 (D$_2$O Solvent) [Supplementary Information (Figure S2)]. [1]H NMR spectra of choline formate (D$_2$O, 600 MHz, $\delta$/ppm relative to TMS): 3.19 (s, 9H, -N-CH$_3$), 3.50, 3.51, 3.52 (t, 2H, -CH$_2$-N-), 4.05 (t, 2H, -O-CH$_2$-), 8.47 (s, 1H, HCOO-), 4.88(D$_2$O Solvent) [Supplementary Information (Figure S3)].

The preparation of deep eutectic solvents (DESs) involved mixing a hydrogen bond acceptor (HBA), namely choline chloride, with hydrogen bond donors (HBDs) such as glycolic acid, acetic acid, and formic acid in a 1:2 molar ratio with continuous stirring. These mixtures were heated at 70 °C until a transparent solution was achieved following the procedure reported elsewhere [21]. The structural (Figure 1). integrity of the DESs was then confirmed by FT−IR analysis. This methodology ensures the formation of homogeneous

and transparent DES solutions, and FT−IR serves as a valuable tool for verifying their chemical composition and stability.

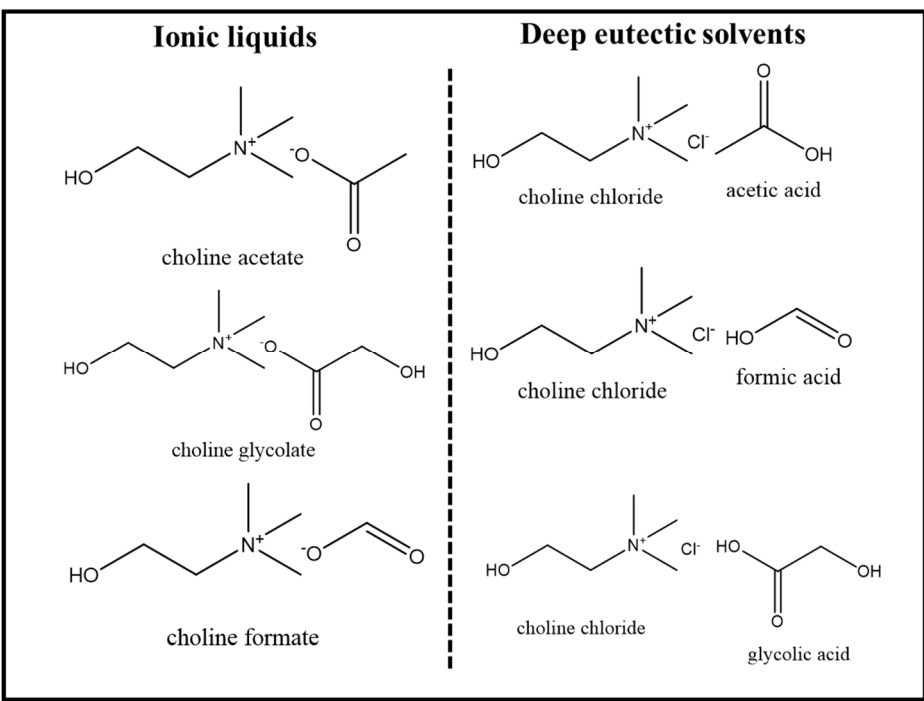

**Figure 1.** Structures of ionic liquids and deep eutectic solvents with similar constituents.

### 2.3. Green Approach: Extraction of Alginate from Sargassum tenerrimum Seaweed

A series of experiments were conducted to extract alginate from seaweed powder using various solvent systems, including ILs and DESs, along with their corresponding aqueous solutions. In **Experiment 1**, a beaker containing 10 g of individual ILs (each IL tested separately) and 500 mg of seaweed powder was employed. In **Experiment 2**, 500 mg of seaweed powder was mixed with ILs containing 10% water. Similarly, in **Experiment 3**, 500 mg of seaweed powder was combined with 10 g of individual DESs. **Experiment 4** involved 500 mg of seaweed powder with DESs containing 10% water. Additionally, a **controlled experiment** was conducted using 500 mg of seaweed powder with 10 g of water. All mixtures were heated to a temperature range of 80–85 °C for 1 h. The resulting mixtures underwent centrifugation, with the separated material at the bottom of the tube. The viscous supernatant was precipitated in isopropanol (IPA) at a 1:3 *v/v* ratio, yielding precipitated choline alginate, which was then dried under vacuum. The seaweed residue remaining after alginate extraction was utilized for the extraction of crude cellulose. Furthermore, the recovered ionic liquid demonstrated its potential for reuse in multiple extraction cycles, highlighting the sustainability and reusability of the chosen solvent system. After the extraction process, approximately 67% of the IL was successfully recovered. The residual mixture comprising IPA, IL, and water, which remained after the precipitation of choline alginate, was subjected to rotary evaporation. During this evaporation step, both IPA and water were removed, resulting in a concentrated IL solution. It is worth noting that this concentrated IL solution is not 30% hydrated. Additionally, the IPA utilized in the extraction process was effectively recovered and is available for reuse in subsequent experiments.

### 2.4. Extraction of Cellulose from Seaweed Residue

Cellulose extraction followed a modified procedure based on [32], as described by [33]. In this process, the residual seaweed biomass remaining after choline alginate extraction underwent several treatments. Initially, it was soaked in $H_2O_2$ for two days for bleaching.

The bleached biomass was then rinsed until it was neutral and treated with a 0.5 M NaOH solution, before being heated overnight at 60 °C. After neutralization, filtration, and drying at room temperature (32 °C), the dried product we obtained was resuspended in a 5% *v/v* hydrochloric acid (HCl) solution. Subsequently, this suspension underwent boiling, and the resulting slurry was allowed to stand overnight. This was followed by several water washings to remove any excess acid. The produced cellulose underwent structural confirmation through FT-IR spectroscopy. The spectrum exhibited a peak at 3340 cm$^{-1}$, corresponding to hydroxyl group stretching, indicative of intramolecular and intermolecular hydrogen bonds. In the FT-IR spectra, the peak at approximately 1630 cm$^{-1}$ is attributed to the stretching of bound H$_2$O. Similarly, the peak at 1425 cm$^{-1}$ corresponds to C-H bending, commonly observed in the range of 1420–1422 cm$^{-1}$ [27]. Distinctive vibrations of the glucose unit's C-H group were observed at 2989 cm$^{-1}$ (stretching) and 1367 cm$^{-1}$ (deformation). The β-glycosidic linkage connecting the glucose units was identified through the absorption band observed at 898 cm$^{-1}$, whereas the signal at 1056 cm$^{-1}$ indicated the presence of the –C–O– group in secondary alcohols and ether functionalities within the cellulose chain backbone [Supplementary Information (Figure S8)].

### 2.5. Conventional Approach: Extraction of Alginate from Sargassum tenerrimum

The extraction of sodium alginate was conducted using a method previously outlined by the authors of [34] in 2020. In this routine procedure, 30 g of seaweed underwent meticulous washing to remove epiphytes and sand. The extraction process involved two primary treatments. Firstly, the seaweed was treated with a 5% HCl solution (1:10 *w/v*) at room temperature for 12 h, resulting in an acidic seaweed residue. This residue was then thoroughly washed with water until a neutral pH of 7 was reached. Subsequently, the washed residue underwent alkali treatment with a 5% Na$_2$CO$_3$ solution for 12 h at room temperature. The resulting reaction mixture was filtered, separating a viscous alkaline extract and seaweed residue. The viscous filtrate was further processed by pouring it into isopropanol (IPA) in a 1:3 *v/v* ratio, followed by vacuum drying. This process yielded sodium alginate with a 22.5% extraction efficiency relative to the seaweed used. Overall, the methodology successfully isolated sodium alginate from the seaweed biomass, demonstrating its effectiveness for practical applications.

### 2.6. Molecular Docking

A molecular docking analysis involving alginate and ligands, specifically IL ions, and corresponding Deep ep eutectic solvents (HBA and HBD), was conducted using the AutoDockVina 1.1.2 program [35]. The molecular structure of alginate was built using Discovery Studio, v20 (Accelrys, San Diego, CA, USA), and AutoDockTools (ADT) [36] was utilized to generate alginate input files in the .pdbqt format for subsequent use in the molecular docking analysis. The 3D atomic coordinates of ligands, comprising IL ions and corresponding deep eutectic solvents (HBA and HBD), were calculated through using Discovery Studio. AutoDockTools (ADT) was employed to generate the rigid root of the ligands, with the activation of all potential rotatable bonds defined as active by torsions.

Concerning the docking analysis, the grid center for alginate was established at coordinates 48.311 Å × 14.939 Å × −7.567 Å along the x-, y-, and z-axes, respectively. The grid dimensions were set to 50 Å × 50 Å × 50 Å. The binding model exhibiting the lowest binding free energy was derived using specific configurations: exhaustiveness = 100, num_modes = 10, and seed = 100 for each ligand. This molecular docking approach provides valuable insights into the potential interactions between alginate and the tested ligands, aiding in the understanding of their binding affinities and modes.

### 2.7. Characterization

Various analytical techniques were employed to characterize the properties and structure (Figures 2 and 3) of choline alginate and sodium alginate. The $^{13}$C and $^{1}$H NMR spectra were recorded using a 600 MHz JEOL ECZ600R spectrometer equipped with a

ROYAL probe. For the $^1$H NMR studies, 1% solutions of choline alginate and sodium alginate were prepared in D$_2$O (99.9%, 1 mL), with tetramethylsilane (TMS) serving as an internal reference for calibrating chemical shifts. The NMR spectra of ionic liquids (ILs) and recovered ILs were captured under the same conditions.

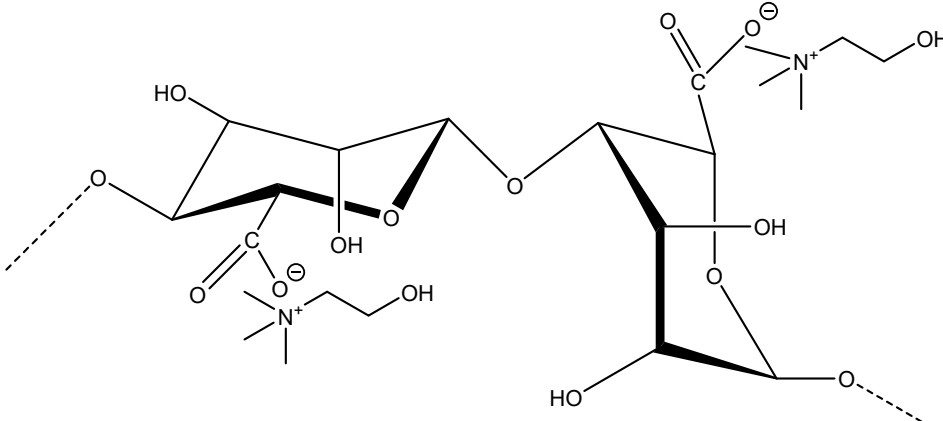

**Figure 2.** Structure of choline alginate.

**Figure 3.** Repeating unit of glucuronic acid and mannuronic acid in choline alginate.

Fourier-transform infrared (FT-IR) analysis spanning the 4000–650 cm$^{-1}$ range was performed with KBr pellets using an Agilent Technologies FTIR-630 spectrometer. Additionally, secondary derivatives of FT-IR spectra within the range of 800–1250 cm$^{-1}$ were computed using Origin software version 2019b (9.65).

Rheological tests were conducted using the Anton PaarPhysica MCR301 Rheometer (Ashland, VA, USA) equipped with a PP50/P-PTD200 parallel plate geometry (49.971 mm diameter; 0.3 mm gap). Steady shear viscosities were measured in the range of 1 to 1000 s$^{-1}$ at 25 °C. The dependent storage modulus (G') and loss modulus (G'') were determined at an amplitude of 0.1%, a frequency of 0.01 Hz, and a time of up to 1800 s by employing Rheoplus/32 V3.31 software.

A differential scanning calorimetric (DSC) analysis was conducted using a NETZSCH DSC 204F1 Phoenix system, employing a heating rate of 10 °C min$^{-1}$, ranging from room temperature to 250 °C. Samples weighing between 10 and 15 mg were utilized for these measurements.

Thermogravimetric analysis (TGA) was conducted using a NETZSCH TG 209F1 Libra Instrument to assess the thermal stability of the materials. The analysis involved a

heating rate of 10 °C min$^{-1}$ from room temperature to 600 °C in a nitrogen atmosphere. These analytical techniques collectively provided a comprehensive understanding of the structural, thermal, and rheological properties of the investigated materials.

## 3. Results and Discussion

The extraction process began with finely ground *Sargassum tenerrimum* powder, which was treated separately and washed with water to eliminate epiphytes, sand, and soluble impurities. Subsequently, the seaweed powder was immersed in ionic liquids (ILs), deep eutectic solvents (DESs), and their hydrated mixtures, as detailed in the Materials and Methods section, maintaining a ratio of 1:10 ($W/V$) between seaweed and the solvent system. These mixtures underwent heating at 80 °C–85 °C for 1 h. After centrifugation, the resulting seaweed residue was set aside for drying. The viscous supernatant obtained was introduced into isopropanol (IPA) in a 1:3 $v/v$ ratio, leading to the precipitation of choline alginate. The residue, after thorough rinsing with IPA, underwent vacuum drying. Choline alginate was also extracted using 10% hydrated DESs, 10% hydrated ionic liquid, and pure water (considered as a control) through the traditional method. This comprehensive approach allowed for a comparative analysis of the physicochemical properties.

The FT-IR analysis of the precipitate revealed the presence of traces of the polysaccharide, providing insights into the structural composition of the extracted choline alginate. This methodology showcases a systematic and versatile approach to alginate extraction, allowing for comparisons with traditional methods and alternative solvents for a thorough understanding of physicochemical characteristics.

We explored the process of extracting choline alginate from *Sargassum tenerrimum* seaweed through the innovative use of ionic liquids (Figure 4). This visual guide showcases the sustainable and efficient steps involved in extracting the seaweed's valuable polysaccharides, highlighting the eco-friendly aspects of this method. Highlighting the potential of ionic liquids paves the way for greener and more sustainable seaweed extraction processes.

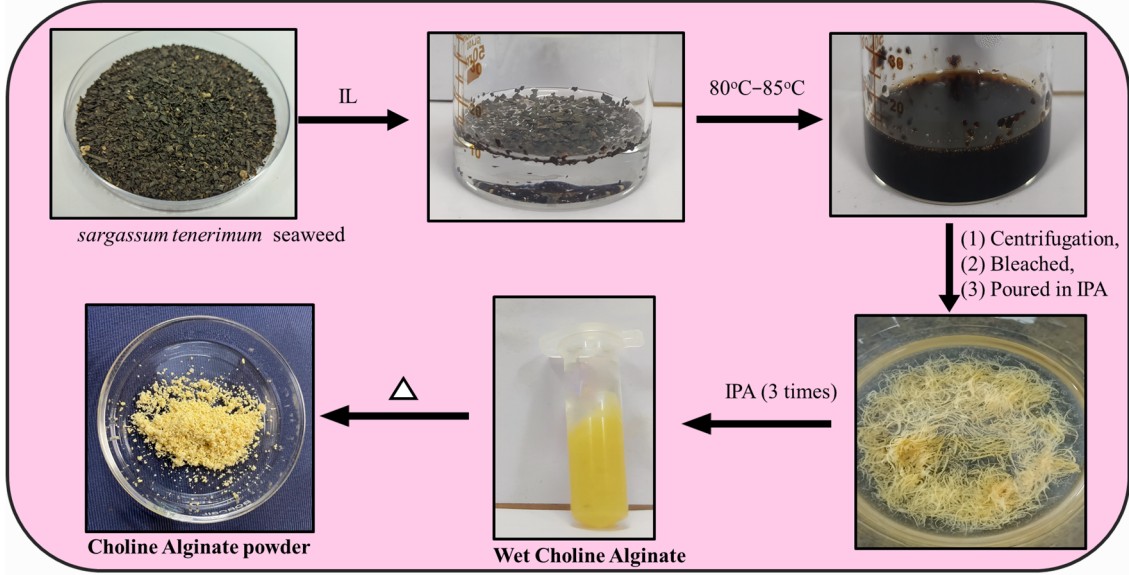

**Figure 4.** Photographic demonstration for the extraction of choline alginate from *Sargassum tenerrimum* seaweed using ionic liquid.

Table 1 highlights a significant increase in choline alginate yields when utilizing hydrated ILs for extraction compared to pure ILs, water, and traditional methods. The highest yield was achieved with 10% hydrated choline glycolate as the extraction solvent, while the yields were notably lower, approaching negligible levels, with deep eutectic solvents (DESs) and hydrated DESs. Comparing pure ILs to their hydrated counterparts, almost consistent yield gains were observed in the hydrated ILs. The extraction of choline alginate

from *Sargassum tenerrimum* involves breaking down the seaweed's cell wall, facilitating interaction with the solvents. Notably, the yield was substantially higher with hydrated choline glycolate compared to traditional extraction methods. Through experimentation with various hydrated ratios, optimal conditions for extraction were identified, particularly with choline glycolate. Further refinements in the extraction process, guided by the superior yields observed in hydrated choline glycolate, led to the maximization of the choline alginate yield. Table 2 presents the analysis results based on hydration ratios of 30%, 50%, 70%, and 90%, yielding choline alginate. Notably, the 40% hydration ratio demonstrated a yield close to 30%, prompting further investigation. Multiple batches using a 30% aqueous solution of choline glycolate consistently yielded 54.09%, significantly surpassing conventional methods.

**Table 1.** Choline alginate yields relative to dried seaweed across various solvent systems.

| Entry | Solvent System | Yield % (±S.D.) | Sample Code |
|-------|---------------|-----------------|-------------|
| 1. | Choline acetate | 5.8166 ± 2.7768 | **A** |
| 2. | Choline formate | 14.847 ± 6.4619 | **B** |
| 3. | Choline glycolate | 19.55 ± 2.2307 | **C** |
| 4. | 10% hydrated choline acetate | 7.2033 ± 2.9859 | **D** |
| 5. | 10% hydrated choline formate | 15.13 ± 7.4486 | **E** |
| **6.** | **10% hydrated choline glycolate** | **25.96 ± 3.4530** | **F** |
| 7. | Choline chloride/acetic acid | Minimal precipitate detected | **G** |
| 8. | Choline chloride/formic acid | Minimal precipitate detected | **H** |
| 9. | Choline chloride/glycolic acid | Minimal precipitate detected | **I** |
| 10. | 10% hydrated choline chloride/acetic acid | Minimal precipitate detected | **J** |
| 11. | 10% hydrated choline chloride/formic acid | Minimal precipitate detected | **K** |
| 12. | 10% hydrated choline chloride/glycolic acid | Minimal precipitate detected | **L** |
| 13. | Water | 7.71 ± 1.638 | **M** |
| 14. | Conventional method * | 22.5% | **NA** |

* [34]; NA = not applicable.

**Table 2.** Optimization of hydrated ratio.

| Sr. No. | Solvent System | Yield % |
|---------|---------------|---------|
| 1. | 30% hydrated choline glycolate | 54.09 ± 2.72 |
| 2. | 40% hydrated choline glycolate | 30.00 ± 2.12 |
| 3. | 50% hydrated choline glycolate | 26.56 ± 3.45 |
| 4. | 70% hydrated choline glycolate | 18.94 ± 2.56 |
| 5. | 90% hydrated choline glycolate | 9.64 ± 1.48 |

In contrast, when deep eutectic solvents (DESs) were employed as the extraction solvents, trace amounts of precipitation were observed, both with the pure DES and its hydrated mixture. These findings underscore the efficacy of 30% hydrated choline glycolate in achieving higher yields compared to DES, emphasizing its potential as a superior solvent system for choline alginate extraction.

The extraction ability of ILs and DESs was evaluated for alginates from brown seaweed, and the results obtained reveal that ILs/DESs' chemical structure induces a significant impact on alginate extraction. Therefore, a molecular docking analysis was carried out to identify the interactions between ILs/DESs (IL ions and the corresponding deep eutectic solvents HBA and HBD) and alginate. Docking affinities (kcal/mol), as well as information on the type of interaction and the geometry distance (A°) for all the studied ligands can be found in the Supplementary Information (Table S1).

According to the molecular docking results, the docking affinity of IL ions follows the following rank: [Glycolate] −> [Ch]+ > [Acetate] − > [Formate] − > Cl⁻. All IL ions (cation and anions) display hydrogen bondability with alginate's structure. However, IL cation ([Ch]⁺) also shows electrostatic interaction from a nitrogen atom (positively charged) to

an alginate carbonyl group (COOH). The hydrogen bondability of the tested ILs increased following the order of [Ch][Acetate] < [Ch][Formate] < [Ch][Glycolate]. The experimental results obtained show a similar trend of extraction yields, which allows us to describe hydrogen bondability as a major driving force for alginate extraction (higher hydrogen bondability to alginate structure leads to higher extraction yields). In addition, all IL anions interact only with alginate hydroxyl groups, which allows for the interaction of IL cations with alginate carbonyl groups. On the other hand, the IL counterpart DESs with acetic, formic, and glycolic acids were not able to extract alginate. The docking affinity of DES HBDs decreases following the rank of glycolic acid < acetic acid < formic acid. All HBDs exhibit higher hydrogen bondability in comparison with their ILs analog anions. Thus, they were expected to be more effective in alginate extraction. However, the stability of DESs in aqueous solutions at lower concentrations was the main reason for the absence of results at an experimental level. The amount of water could promote the decrease in interactions between the DESs of HBA and HBD, and both components act as individual compounds in water solution and not as a solvent, making it difficult to predict the impact of DESs on the extraction procedure [37] [Supplementary Information (Figure S9)].

To find out the quality and structural integrity of choline alginate extracted through an innovative, environmentally friendly approach and a conventional method, FT-IR spectra were recorded for both samples (Figure 5). In this analysis, choline alginate extracted from a 30% hydrated system was utilized. In Figure 5, the distinctive blocks for M block and G block (uronic acid) manifest at 808 cm$^{-1}$ and 944 cm$^{-1}$, respectively. These spectral features align with those of Na-Alg, as reported in [29]. The peaks at 1479 cm$^{-1}$, 1082 cm$^{-1}$, and 1054 cm$^{-1}$ are attributed to the bending vibration of C-O-H, stretching vibration of C-O, and stretching vibration of C-N in the [ch]+ moiety of choline chloride, respectively. These findings are consistent with the corresponding spectrum of choline chloride. Preliminary data indicate the presence of [Alg]$^{-}$ and [ch]$^{+}$ in the structure of the extracted product, supporting its chemical composition [38]. Furthermore, the structural integrity of ch-Alg was confirmed by $^1$H NMR and $^{13}$C NMR spectroscopy as well. The $^1$H NMR spectra of choline chloride showed peaks at 3.21 (s, 9H, -NCH$_3$), 3.52, 3.53, 3.54 (t, 2H, -CH$_2$-N-), 4.06, 4.07, and 4.08 (t, 2H, -O-CH$_2$) [Supplementary Information (Figure S10)]. $^1$HNMR peaks for sodium alginate were observed at 5.05 [CH-1 of M&G], 4.64 [1H, CH-5 of G], 4.46 [1H, CH-4 of G], 4.19 [1H, CH-3 of G], 4.03 [CH-2 of G], 3.99 [CH-2 of M], 3.89 [CH-5 of M], and 3.75 [CH4 of M], as reported earlier [29] [Supplementary Information (Figure S11)]. $^1$H NMR peaks were also observed in the choline alginate samples at 5.08 [CH-1 of M&G], 4.65 [1H, CH-5 of G], 4.54 [1H, CH-4 of G], 4.17 [1H, CH-3 of G], 4.02 [CH-2 of G], 3.99 [CH-2 of M], 3.88 [CH-5 of M], and 3.74 [CH-4 of M], which indicated the presence of the G-block and M-block of alginate, and the peaks at 3.20 (s, 9H, -NCH$_3$), 3.52 (t, 2H, -CH$_2$-N-), and 4.06 (t, 2H, -O-CH$_2$), as shown in Figure 6, indicated the presence of choline moieties in the product [29]. Further, in the $^{13}$C NMR spectra of the sample, distinct peaks corresponding to various moieties are evident. The choline moiety exhibited specific peaks, e.g., three methyl groups (CH$_3$) are represented by a sharp peak at 53.93 ppm, while two methylene groups (CH$_2$) were observed at 55.66 ppm and 67.48 ppm. Additionally, the alginate moiety contributes a peak corresponding to the C-6 carbon at 179.90 ppm (Figure 7).

The thermal stability of choline alginate was investigated using TGA and DSC analysis. Na−Alg and Ch−Alg exhibited initial decomposition temperatures of 224 °C and 180 °C, respectively [Supplementary Information (Figure S12)]. This suggests that both polysaccharides maintain their thermal stability until 180 °C. The DSC analysis results were consistent with the TGA data, revealing no phase transformation point before the decomposition point [Supplementary Information (Figures S13 and S14)]. Both analyses collectively demonstrate the broad thermal stability range of choline alginate, spanning from ambient temperature to 180 °C.

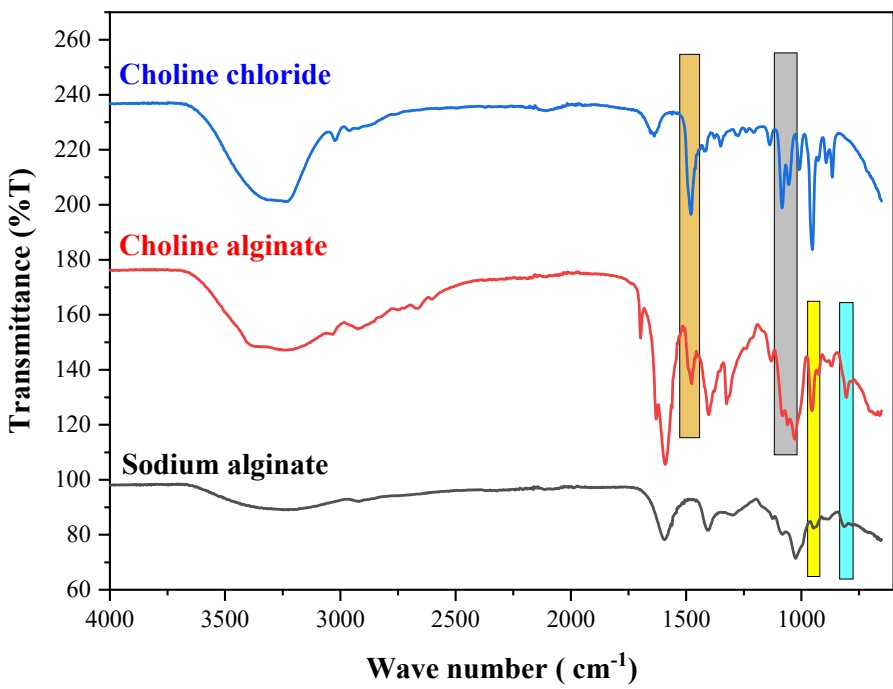

**Figure 5.** FT−IR spectra of choline alginate, choline chloride, and sodium alginate.

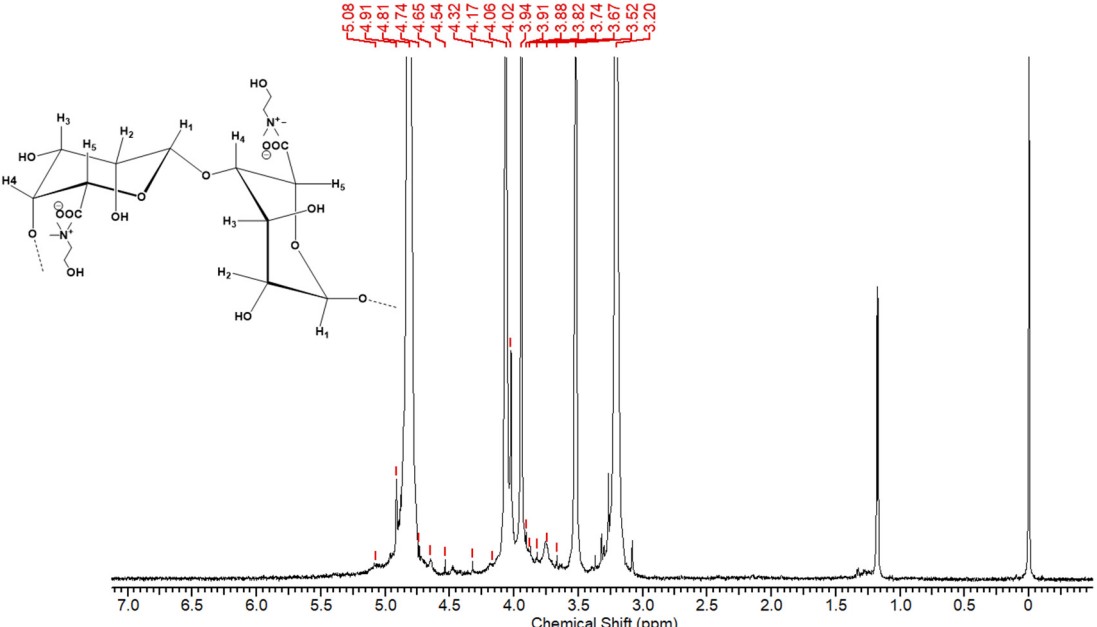

**Figure 6.** $^1$H NMR spectra of choline alginate.

As alginates are commonly employed as gelling and thickening agents, a rheological investigation was conducted to assess the behavior of alginate samples extracted through conventional methods and those extracted using an ionic liquid system, allowing for a comparison of their rheological characteristics. Various concentrations (1%, 5%, 10%) of solutions from both extracted alginates were analyzed, with the graph illustrating the results of a 10% viscous solution of the alginate. As depicted in Figure 8, the flow property of choline alginate is higher than that of the alginate extracted using the conventional method (Na−Alg). This means that Chol−Alg has a better pseudoplastic or shear thinning nature than Na-Alg.

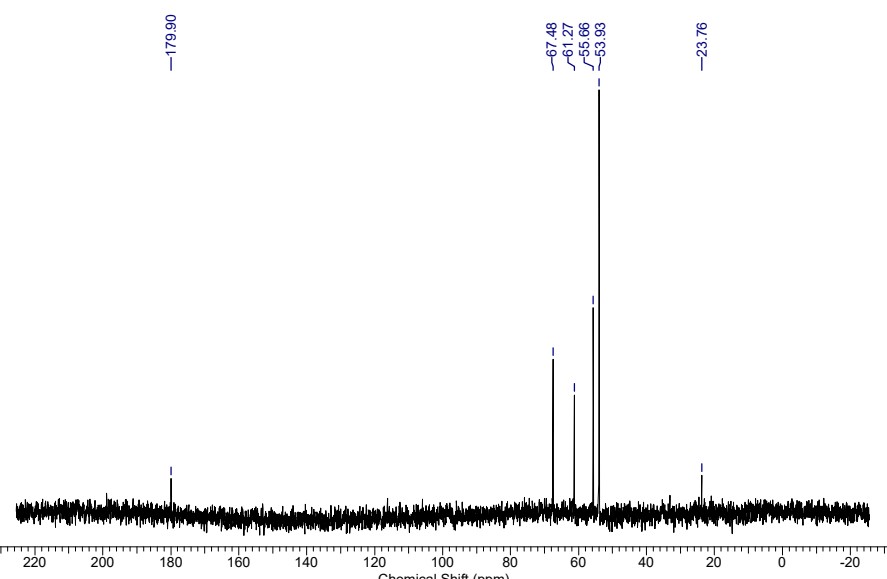

**Figure 7.** $^{13}C$ NMR spectra of choline alginate.

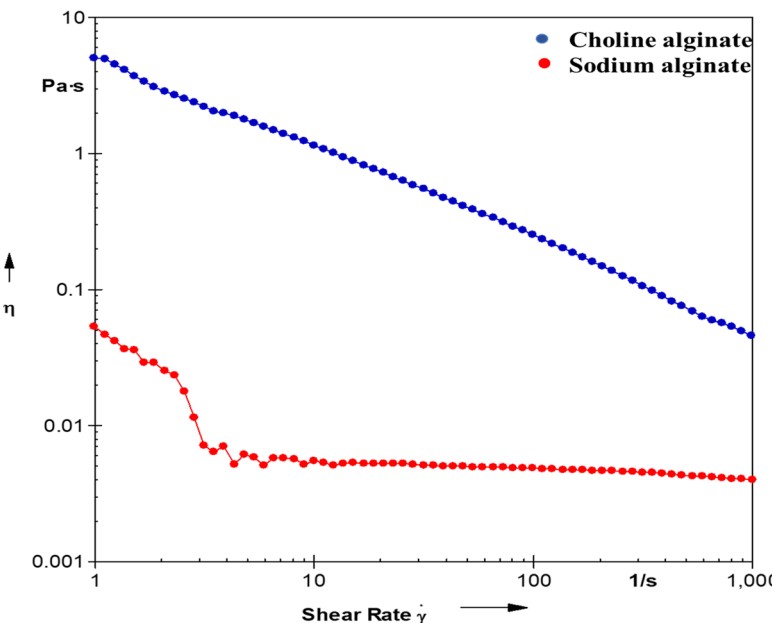

**Figure 8.** Steady shear viscosity of choline alginate and sodium alginate.

The time-dependent viscoelastic behavior reveals significant differences, with the magnitudes of $G'$ and $G''$ being highest for choline alginate extracted using an aqueous solution of ionic liquids compared to sodium alginate extracted using the conventional method. Notably, the difference between $G'$ and $G''$ is minimal for sodium alginate, as depicted in Figure 9. The flow behavior and viscoelasticity profiles collectively suggest that choline alginate exhibits superior viscoelastic behavior compared to its counterpart, which was extracted using traditional approaches.

The crossover of $G'$ and $G''$ was observed in all alginate solutions with increasing frequency, indicating the frequency-dependent characteristic of weak gels (Supporting Figure S15). However, the frequency at which the crossover occurred varied among the gels. Interestingly, the presence of additional flow components in the gels caused them to act like viscous liquids at lower frequencies, with the loss modulus predominating over the storage modulus across a wide frequency range. Specifically, the sample of sodium alginate displayed liquid behavior, while choline alginate exhibited viscous liquid behavior. The

rheological investigation underscores that choline alginate possesses superior viscosity and flow behavior compared to sodium alginate.

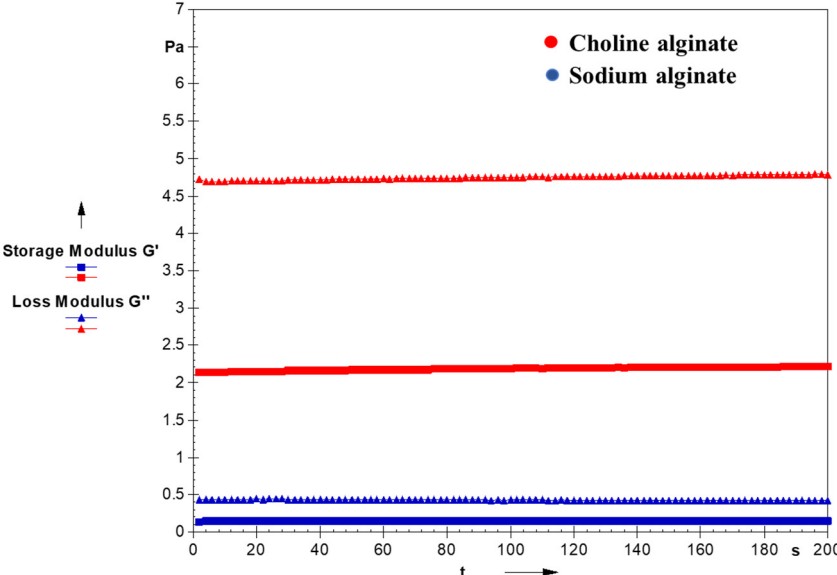

**Figure 9.** Time-dependent viscoelasticity behavior of choline alginate and sodium alginate.

## 4. Conclusions

In conclusion, this study meticulously investigated the efficacy of various techniques for extracting alginate from *Sargassum tenerrimum*, including ionic liquids (ILs), deep eutectic solvents (DESs), and their respective aqueous solutions. Through comprehensive analysis, it was determined that a 30% aqueous solution of IL emerged as a highly promising alternative solvent, surpassing conventional methods in terms of both the quality and efficiency of alginate extraction.

The comparison between these techniques revealed critical insights into their respective advantages and limitations. Notably, the greener approach, characterized by the utilization of modern technology and minimal organic solvents, exhibited superior environmental sustainability, method reproducibility, and a reduced extraction time compared to the conventional methods. Moreover, the use of inexpensive and nontoxic materials further enhanced the cost-effectiveness and minimized the toxicity concerns associated with traditional extraction techniques.

Furthermore, the greener approach showcased the ability to recover solvents, thereby reducing waste and environmental pollution. This contrasts starkly with the conventional method, which produces hazardous by-products such as alkali and acid effluents and yields lower overall efficiency. Importantly, any solid residual material remaining after the extraction process was successfully converted into cellulose. This innovative feature of the greener extraction method ensures the complete utilization of the raw material, resulting in zero water discharge and further enhancing the sustainability of the process. Overall, the findings underscore the paramount importance of transitioning towards greener extraction methods to promote sustainability and efficiency in alginate extraction processes. By prioritizing environmental responsibility and embracing innovative technologies, researchers can contribute to the advancement of sustainable practices in polysaccharide extraction, thereby ensuring a healthier and more environmentally conscious future.

**Supplementary Materials:** The following supporting information can be downloaded at https://www.mdpi.com/article/10.3390/suschem5020009/s1. Schemes S1–S3: Reaction Scheme for the synthesis of ionic liquids and deep eutectic solvents, Figures S1–S8: NMR and FT-IR spectra of the ionic liquids and deep eutectic solvents. Table S1 and Figure S9: Docking affinity and pose of the interactions. Figures S12 and S13: DSC, TGA of choline and sodium alginate. Figure S14: DSC

curve of Choline alginate. Figure S15: Frequency dependence of G′ and G″ for alginate prepared conventional and greener approach.

**Author Contributions:** All authors actively contributed to the study. K.P was responsible for the study's conceptualization and design. K.M. and M.M.P. were responsible for material preparation, data collection, and analysis. The initial draft of the manuscript was composed by K.M., and the manuscript was subsequently reviewed and edited by K.P. All authors have read and agreed to the published version of the manuscript.

**Funding:** The authors are thankful for the funding they received from the Council of Scientific and Industrial Research, New Delhi, India (HCPOO24).

**Institutional Review Board Statement:** Not applicable.

**Informed Consent Statement:** Not applicable.

**Data Availability Statement:** No new data were created or analyzed in this study. Data sharing is not applicable to this article.

**Acknowledgments:** K.P. and K.M. express gratitude to CSIR, New Delhi, for their financial support (HCP0024) and CSIR-CSMCRI. The authors also acknowledge the Analytical Discipline and Centralized Instrumental Facilities of the Institute for providing the necessary instrumentation facilities for this work. M.M.P. acknowledges the financial support of FCT, Portugal, within the project DOI: 10.54499/UIDB/00102/2020 (base funding) and DOI: 10.54499/UIDP/00102/2020 (programmatic funding).

**Conflicts of Interest:** The authors declare no conflicts of interest.

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
