# Peer review of "Aqueous Solution of Ionic Liquid Is an Efficient Substituting Solvent System for the Extraction of Alginate from Sargassum tenerrimum"

_2673-4079, doi:10.3390/suschem5020009_

Round 1

Reviewer 1 Report

Comments and Suggestions for Authors

The manuscript is dedicated to the extraction of alginate from Sargassum tenerrimum using aqueous solution of ionic liquids. The paper is quite well written with an adequate format for Sustainability with a subject appropriated to the scope of the journal. The references have to be also revised for a care of uniformity and the quality of the GA and of some figures or schemes has to be largely improved. Many typing errors have to be corrected too.

The introduction is devoted to alginate (composition, uses and extractions) and to ionic liquids and DES. The corresponding literature is recent for most references.

Next, an extraction diagram could have been added to explain more clearly the process and solvent recycling.

Then, the systematic study (ionic liquids with various anions, DES and water %) is well conducted. The conclusions resulting from docking, IR or NMR studies help to understand the experimental results, but here too, everything is rather confused, and more precise or orderly explanations would have added value to this manuscript, especially by comparison with extractions involving classical organic. DRX experiments on recovered cellulose have to be also realized.

In conclusion, in this form, I can’t accept this manuscript for a publication in Sustainability.

Comments on the Quality of English Language

See the previous report.

Author Response

Reviewer 1:

Comments- The manuscript is dedicated to the extraction of alginate from Sargassum tenerrimum using aqueous solution of ionic liquids. The paper is quite well written with an adequate format for Sustainability with a subject appropriated to the scope of the journal. The references have to be also revised for a care of uniformity and the quality of the GA and of some figures or schemes has to be largely improved. Many typing errors have to be corrected too.

Author’s response - The manuscript has been revised by the authors based on suggestions. All references have been updated, and few figures have been replaced with higher-quality versions. The graphical abstract has been improved for better clarity, and typing errors have been corrected to enhance readability.

Comments-extraction diagram could have been added to explain more clearly the process and solvent recycling.

Author’s response -The extraction diagram in the manuscript has been revised to include detailed information about the solvent recycling process. The updated diagram now provides a clear visual representation of the extraction process, highlighting the stages where solvent recycling occurs.

Additionally, a section has been incorporated into the manuscript that elaborates on the solvent recycling process, providing a comprehensive explanation to ensure readers understand the importance and implementation of solvent recycling in the extraction procedure.

Comments - Then, the systematic study (ionic liquids with various anions, DES and water %) is well conducted. The conclusions resulting from docking, IR or NMR studies help to understand the experimental results, but here too, everything is rather confused, and more precise or orderly explanations would have added value to this manuscript, especially by comparison with extractions involving classical organic. DRX experiments on recovered cellulose have to be also realized.

Author’s response - A comparative analysis between the extraction method using ionic liquids or Deep Eutectic Solvents (DES) and classical organic solvent-based methods has already been included in the graphical part of the manuscript. Furthermore, the DRX experiment data on the recovered cellulose is also mentioned in the manuscript.

Below table is provided for better explanation of above. This is now discussed in the revised manuscript.

Conventional approach

Greener approach

Ø  organic solvents are used which is hazardous and cause harmful effects on the environment and health impacts

Ø  Time consuming,

Ø  Costly

Ø  toxic materials used,

Ø   unable to recovered solvent used for extraction

Ø  All alkali, and acid effluent are hazardous by-product remaining

Ø  Near 30 % yield obtained for sodium alginate

ü  where fewer or no organic solvents are used to minimize environmental and health impacts

ü  Faster

ü  Cost effective

ü  Emphasised to use bio-based chemicals 

ü  ambient conditions, and

ü  Availability of raw materials.

ü  Recovery the solvents by recycling

ü  No by-product observed

ü  The yield enhanced to 54 %

Reviewer 2 Report

Comments and Suggestions for Authors

The manuscript by Moradiya, K. et al. explores the potential of ionic liquids (ILs), deep eutectic solvents (DES), and their aqueous solutions for selectively extracting alginate from algal biomass. The use of environmentally friendly solvents for processing and fractionation of plant biomass is a promising direction for the development of biorefining technologies, aligning with the principles of green chemistry. The study takes a comprehensive approach to the extraction of alginate from algal biomass using ILs, DES, and their aqueous solutions. The obtained preparations are characterized using modern instrumental methods of analysis. The focus is on determining the structural features of the extracted alginates. The manuscript may be of interest to the audience of Sustainable Chemistry and is worthy of publication. However, minor corrections are required.

Some recommendations and questions are presented below:

Line 93 - It is recommended to highlight the results of other authors on fractionation of algal biomass using DES and ILs and characterization of the resulting preparations.

Line 180 - It is suggested to clarify whether the authors detected lignin or if the absorption bands are due to the presence of polyphenols/phlorotannins?

Line 302 - Is it possible that the use of solvents diluted with more than 30% water does not involve a binary mixture of IL and water, but rather a solution of salt and acid in water? Lastly, would using a 30% aqueous solution of glycolic acid yield similar results?

Author Response

Reviewer 2:

Comments: Line 93 - It is recommended to highlight the results of other authors on fractionation of algal biomass using DES and ILs and characterization of the resulting preparations

Author response -The recommendation to highlight the results of other authors on the fractionation of algal biomass using Deep Eutectic Solvents (DES) and Ionic Liquids (ILs), as well as the characterization of the resulting preparations, has been addressed.

“ Studies have shown that hydrated DES exhibits superior extraction efficiency for carra-geenan from Kappaphycus, emphasizing the potential of DES as an effective solvent sys-tem for biomass extraction and highlighting the importance of considering solvent hydra-tion in optimization strategies for biomass fractionation[28].previous studies have em-ployed Ionic Liquids (ILs), such as Tetramethylammonium hydroxide (TMAH), for the ex-traction of alginic acid from seaweed polysaccharides. The extraction process involves an acid treatment followed by the use of TMAH, highlighting the versatility of ILs in poly-saccharide extraction processes[29]”

Comments: Line 180 - It is suggested to clarify whether the authors detected lignin or if the absorption bands are due to the presence of polyphenols/phlorotannins?

Author’s response -Distinguishing between lignin and polyphenols/polytannins in FT-IR spectra can be challenging due to the overlapping functional groups and similar peak positions. Additionally, lignin is likely not present in cellulose extracted from seaweed, further complicating the interpretation of the observed peaks.

“The FT-IR spectra, the peak at approximately 1630 cm-1 is attributed to the stretching of bound H2O, Similarly, the peak at 1425 cm-1 corresponds to C-H bending, commonly observed in the range of 1420–1422 cm-1The authors have corrected line no. 180 and highlighted the information accordingly in the manuscript to reflect this interpretation accurately.”

Comments: Line 302 - Is it possible that the use of solvents diluted with more than 30% water does not involve a binary mixture of IL and water, but rather a solution of salt and acid in water? Lastly, would using a 30% aqueous solution of glycolic acid yield similar results?

Author’s response – Throughout the designed solvent systems, there is no binary mixture observed due to the water solubility of the ionic liquid (IL). This characteristic ensures that the IL remains in a dissolved state rather than forming a distinct binary phase with water.

The use of a 30% aqueous solution of glycolic acid presents unique challenges. The interactions highlighted in the manuscript show that the IL cation ([Ch]+) exhibits electrostatic interactions with an alginate carbonyl group (COOH). The choline moiety presence is crucial for enhancing extraction due to its unique properties. Furthermore, the prepared ILs possess several advantageous properties, including tunability with targeted compounds. Choline exhibits a synergistic effect, emphasizing its importance in the extraction process. If solely glycolic acid is used without the choline component, questions arise regarding the recovery of the IL after extraction, given the absence of the choline component.

Reviewer 3 Report

Comments and Suggestions for Authors

This manuscript showed the potential of ionic liquids for the extraction of  biopolymer from seaweed. The authors have successfully isolated Alginate salts from Sargassum tenerrimum seaweed with hydrated ionic liquid and compared their efficiency with DES and conventional solvent systems. The results showed that hydrated ionic liquids performed better than non-hydrated ionic liquids and DES. The experiments are well planned and the analysis was performed with great care. 

Some suggestions to the authors for an improvement:

It is better to provide reference for the first two lines since it introduce the importance of alignate.

The first paragraph has contain repetitive informations on the application of alignate, so it can be modified for readability. This applies to the second paragraph also.

The first three paragraph should be modified for improvement without repeating the same informations.

The reference section should be improved for example the introduction of Ionic liquid just has only one reference (a review) but can be supplement with many other important references in the field.

The discussion based on references no. 8-13 can be supplement with other related extraction to this study.

Line no 78 the authors claiming DES is made up of risk free components but this is misleading as some of the components used in the DES making is actually toxic for example acetamide. So this should be modified.

It could be more interesting if the authors explain the reason behind the choice of these six novel eco-friendly solvents for the extraction.

The paragraph below figure 4. could be improved or should be rewritten as it has flaws. 

Table 1,  Sr. No can be replaced with Entry.

When DES used for extraction there is a precipitation observed as cited in Table 1. By the way what it is, has the authors isolated or identified?

Can the authors be more specific to the line no. 299, which hydrated ratio was used here?

The authors mentioned in line No 164 that the recovered ionic liquid showed potential for reuse but there is no additional information on how much the yield for efficiency. I don't find any other table in the results and discussion part. 

Please provide exact time for the preparation of solvents, as it is unalike from SI to Experimental part.

Please provide solvent name and MHz on the NMR spectra.

Since it is mentioned in the manuscript 13C NMR should be provided.

In addition there are several typo errors can be seen throughout the manuscript, that needs to be corrected.

Additionally, the reading is interrupted by introducing cellulose extraction from the residual seaweed as it has not been introduced before. This has to be mentioned somewhere in the abstract/conclusion sections.

In the synthesis of ionic liquids section (line no 119) metathesis should be written as salt metathesis.

Comments on the Quality of English Language

The quality of english language for this manuscript has to be improved. 

Author Response

Reviewer 3:

Comments- It is better to provide reference for the first two lines since it introduce the importance of alginate.

Author’s response – Reference have been updated.

Comments - The first paragraph has contain repetitive information on the application of alignate, so it can be modified for readability. This applies to the second paragraph also.

The first three paragraph should be modified for improvement without repeating the same informations.
Author’s response –Authors have removed few line from the introduction to address this comments–

The widespread use of alginates across these industries underscores their adaptability and underscores their importance in contributing to sustainable and eco-friendly practices

This versatile compound finds applications in diverse sectors such as healthcare, pharmaceuticals, cosmetics, and food industries

The paragraphs are also modified as suggested.

Comments -The reference section should be improved for example the introduction of Ionic liquid just has only one reference (a review) but can be supplement with many other important references in the field

Author’s response – References have been updated.

Comments -The discussion based on references no. 8-13 can be supplement with other related extraction to this study.

Author’s response- Additional related studies have been identified and incorporated into the discussion section of the manuscript. These new references highlight the importance and use of ionic liquids (ILs) as solvents for the extraction of various compounds across different fields.

The added references enhance the depth and relevance of the discussion, offering readers a more comprehensive understanding of the extraction methods utilizing ILs and their implications in the broader context of the field.

Comments- Line no 78 the authors claiming DES is made up of risk free components but this is misleading as some of the components used in the DES making is actually toxic for example acetamide. So this should be modified.

Author’s response –Thank you for input. The sentence is corrected by author and this reads as

'DESs are composed of two or three components that self-assemble through hydrogen bonds, creating a eutectic with lower melting temperatures. Some components may carry risks.' This revision is highlighted in the manuscript to underscore potential risks associated with certain DES components.

Comments It could be more interesting if the authors explain the reason behind the choice of these six novel eco-friendly solvents for the extraction.

Author’s response –The selection of these six solvents was based on their bio-origin and eco-friendly nature. Unlike many conventional ILs and DESs, these solvents pose fewer environmental risks. The aim was to adopt a more sustainable and greener approach for alginate extraction compared to conventional methods. These solvents offer a less toxic alternative to organic solvents traditionally used for extraction, aligning with a commitment to environmentally responsible practices. Further, due to bio-origin these solvents are safe to discard if not recycled.

Comments The paragraph below figure 4. Could be improved or should be rewritten as it has flaws.

Author’s response –The authors have incorporated the suggested changes.

Comments Table 1, Sr. No can be replaced with Entry.

Author’s response –The authors have made the replacement as suggested.

Comments When DES used for extraction there is a precipitation observed as cited in Table 1. By the way what it is, has the authors isolated or identified?

Author’s response –DES yields are not significant for collection, as indicated in Table 1 where only minimal amounts of precipitation were detected. Attempts to collect this negligible amount did not result in measurable weight upon drying. Therefore, there was no meaningful collection or isolation attempted. Additionally, molecular docking studies reveal that DES are not effective in extracting alginate, supported by specific reasons.

Comments- Can the authors be more specific to the line no. 299, which hydrated ratio was used here?
Author’s response –The authors have provided the requested information.

Comments-The authors mentioned in line No 164 that the recovered ionic liquid showed potential for reuse but there is no additional information on how much the yield for efficiency. I don't find any other table in the results and discussion part

 Authors response –Information about the recovery of ionic liquid (IL) has been added by the authors in the revised manuscript.

Comments- Please provide exact time for the preparation of solvents, as it is unalike from SI to Experimental part.

Authors response –The authors have corrected the timing for the preparation of solvents in the revised manuscript to ensure consistency between the Supplementary Information and the Experimental section.
Comments- Please provide solvent name and MHz on the NMR spectra

Authors response –The solvent used for the analysis is D2O, as mentioned in the manuscript's instrumentation section. The NMR spectra were recorded at 600 MHz. The solvent showed its peak at 4.81 MHz.

Comments- Since it is mentioned in the manuscript 13C NMR should be provided.–

Authors response In the 13C NMR spectra of the sample, distinct peaks corresponding to various moieties are evident. The choline moiety exhibits specific peaks: three methyl groups (CH3) are represented by a sharp peak at 53.93 ppm, while two methylene groups (CH2) are observed at 55.66 ppm and 67.48 ppm. Additionally, the alginate moiety contributes a peak corresponding to the C-6 carbon at 179.90 ppm. These peaks provide valuable information about the structural components of both the choline and alginate moieties in the sample. This is now included in the manuscript

Comments-In addition there are several typo errors can be seen throughout the manuscript, that needs to be corrected.

Authors response – The authors have reviewed the manuscript and corrected the grammatical errors to improve the clarity and accuracy of the document.

Comments- Additionally, the reading is interrupted by introducing cellulose extraction from the residual seaweed as it has not been introduced before. This has to be mentioned somewhere in the abstract/conclusion sections.

Authors response -The information about successful cellulose extraction from residual material has been incorporated into both the abstract and conclusion sections.

Comments- In the synthesis of ionic liquids section (line no 119) metathesis should be written as salt metathesis.
Authors response- The word metathesis is now replaced with acid base neutralization reaction which is more correct.

Round 2

Reviewer 1 Report

Comments and Suggestions for Authors

The reviewer thanks the authors for this well revised version. I can accept now this MS to publication.

Comments on the Quality of English Language

The MS has been carefully revised.